# Matching-to-Sample Task Training of a Killer Whale (*Orcinus orca*)

**DOI:** 10.3390/ani14060821

**Published:** 2024-03-07

**Authors:** Ayumu Santa, Koji Kanda, Tomoya Kako, Momoko Miyajima, Ikuma Adachi

**Affiliations:** 1Center for the Evolutionary Origins of Human Behavior, Kyoto University, Inuyama 484-8506, Japan; 2Port of Nagoya Public Aquarium, Nagoya 455-0033, Japan

**Keywords:** marine mammal cognition, killer whale, animal training, visual perception, object recognition, matching-to-sample task, mirror image discrimination

## Abstract

**Simple Summary:**

“How do animals perceive the world, and what are they thinking about?” To approach these questions, a method where animals face a touch screen and work on various visual tasks was used. There have not been many studies conducted on cetaceans (e.g., dolphins and whales) because the presence of water makes it difficult to use the electronic devices needed to study them. In this study, we aimed to train one captive killer whale to perform visual tasks using a monitor through an underwater window as if a touch panel were used. We started the training using real objects, which were then converted to photographs. Finally, we confirmed that the killer whale became able to perform the task even in monitor presentation. The successful use of this method suggests the possibility of conducting more detailed research on killer whale cognitive abilities, and further comparisons between terrestrial animals could be conducted in the future. This method is also expected to contribute to animal welfare, and it could also be used to propose new exhibitions to introduce the cognitive abilities of animals in zoos and aquariums.

**Abstract:**

Matching-to-sample tasks have been a useful method in visual cognitive studies on non-human animals. The use of touch panels in matching-to-sample tasks has contributed to cognitive studies on terrestrial animals; however, there has been a difficulty in using these devices underwater, which is one of the factors that has slowed the progress of visual studies on underwater animals. Cetaceans (e.g., dolphins and whales) are highly adapted to underwater environments, and further studies on their cognitive abilities are needed to advance our understanding of the interactions between environmental factors and the evolution of cognitive abilities. In this study, we aimed to develop a new experimental method in which a captive killer whale performed a matching-to-sample task using a monitor shown through an underwater window as if a touch panel were used. In order to confirm the usefulness of this method, one simple experiment on mirror image discrimination was conducted, and the pairs with mirror images were shown to be more difficult to identify than the pairs with other normal images. The advantages of using this method include (1) simplicity in the devices and stimuli used in the experiments, (2) appropriate and rigorous experimental control, (3) the possibility of increasing the number of individuals to be tested and interspecies comparisons, and (4) contributions to animal welfare. The use of this method solves some of the problems in previous visual cognitive studies on cetaceans, and it suggests the further possibility of future comparative cognitive studies. It is also expected to contribute to animal welfare in terms of cognitive enrichment, and it could help with the proposal of new exhibition methods in zoos and aquariums.

## 1. Introduction

Matching-to-sample tasks have been a useful method in cognitive studies that examine the perceptual, conceptual, and memory abilities of non-human animals. In recent years, many matching-to-sample tasks have been performed using computer-controlled touch panels. There are several advantages in using these devices [1]. First is the intuitive clarity created by the direct relationship between the stimulus projected on the screen and the animal’s response to that stimulus. Second is the stimulus presentation and precise control of the intertrial interval, as well as the automatic recording of responses from subjects. Third, these devises help to remove the experimenter effect caused by the face-to-face interaction between human experimenters and animal subjects. The usefulness of touch panels has been recognized not only in primates (who can use their fingers in the same manner as humans)—such as chimpanzees [2,3,4], bonobos [5,6], orangutans [7,8,9], gorillas [10,11], gibbons [12], mandrills [13,14], and macaques [15,16]—but also in other species that can perform choosing behaviors via their noses, tongues, and beaks, such as dogs [17], bears [18], rodents [19], and birds [20,21,22]. Visual tasks using touch panels have been one of the useful methods for investigating the cognitive abilities of terrestrial animals.

Cetaceans (e.g., dolphins and whales) are highly adapted to underwater environments. In this environment, sound information is exceedingly conductive and more sustainable than when in air [23]; visual information, on the other hand, is degraded due to the scattering, absorption, and attenuation of light by water [24]. Therefore, cetaceans have been considered as dependent on auditory information, and many studies have focused on their auditory abilities. For instance, cetaceans have developed excellent passive hearing abilities [25]. Furthermore, certain toothed whales have acquired echolocation abilities, which is the use of sound as a biological sonar [26]. On the other hand, despite of their low visual acuity [27] and monochromatic color sense [28], observations from research in the wild include that toothed whales’ vision was effectively used for acquiring information from their surroundings, searching for prey, defending against predators, and communication [29,30,31]. Also, in the cross-modal matching tasks between vision and echolocation in captivity, dolphins could quickly and accurately recognize the objects visually that were inspected by echolocation and vice versa [32,33,34]. In this way, cetaceans have not only acquired excellent auditory abilities but also retain good visual abilities, despite living in the underwater environment where visual information is less dominant. Further studies on the visual cognitive abilities of underwater animals, including cetaceans, are expected to advance our understanding about the interaction between environmental factors and the evolution of cognitive abilities. 

Most cognitive studies examining visual abilities of cetaceans have been conducted in the situation that real objects were presented as the visual stimuli in the air [35,36]. These studies have been based on the assumption that dolphins cannot use echolocation in the air due to the differences in the properties between water and air [37]. However, since bats can use echolocation in the air, the possibility that cetaceans are able to use echolocation for real objects presented in the air cannot be completely ruled out. Some similarities have been reported between the echolocation abilities of dolphins and bats [38], and in fact, a study reported that dolphin performed an echolocation task in air as well as in water [39]. Here, touch panel devices that present only visual stimuli that are impossible to explore by echolocation might appear to be useful to eliminate this possibility. However, the situation of having those devices in direct contact with underwater animals is difficult to achieve because the electro-magnetic grid over the touch panel would be short-circuited by the salt water [40]. There have been a few examples of previous studies using touch panels on a sea lion [41] and dolphins [42], but most of these studies were also conducted in the air, such as in poolside environments. Even though their visual abilities are most frequently used in water, the fact that these experiments have been conducted in air calls into question the validity of these methods for examining the true visual abilities of these underwater mammals. Although there have been studies using submersible underwater touch panels [43,44] and an echolocation visualization system [45], these methods are not easy to apply, from the perspective of the housing and exhibition of aquariums, due to the financial and labor costs of certain specialized devices and the difficulty in submerging objects in pools. In this way, the difficulty of using touch panels in underwater environments has been one of the factors that has slowed the progress of visual studies on underwater animals.

Another problem with cognitive studies on cetaceans is that the target species have been mainly limited to bottlenose dolphins (of the genus *Tursiops*). One major reason for this limitation is that the number of bottlenose dolphins kept in aquariums is much larger than other cetaceans and that the training methods used for cognitive experiments on these dolphins via operant conditioning have been established over many years. However, it must be noted that cetaceans include about 90 species, which vary in body size, habitat, diet, life cycle, and social structure. In the current study, the killer whale (*Orcinus orca*) was chosen as a target species. Killer whales have the largest body size of Delphinidae, and they are present all over the world’s oceans [46]. As a distinguishing characteristic of killer whales from other cetaceans, some of them prey not only on fish but also on other marine mammals like dolphins and seals [47]. These prey mammals have a sensitivity to sound frequencies beyond 30 kHz, which overlap with the range of killer whales’ echolocation [48,49,50]. Therefore, it is better for killer whales to refrain from using echolocation when they hunt marine mammals, so that their prey does not notice them from a distance. In fact, previous field studies have reported that mammal-eating killer whales suppress echolocation sounds [51] and communication calls [52] during hunting. They appear to use not only their auditory abilities but also their vision in prey detection [51,53,54,55]. This ecological constraint suggests the possibility that killer whales retain better visual abilities than other cetaceans. However, the small number of research studies to examine the visual cognitive abilities of killer whales has not allowed comparisons with other cetaceans. 

In this study, we trained one captive killer whale to learn the rule of a delayed identity matching-to-sample task. In this task, the subject was first shown a sample stimulus, and after it was disappeared, two comparison stimuli were presented. The subject received positive feedback if it chose one of the comparison stimuli that was the same as the sample. At the beginning of the training, three familiar real objects were used to help the subject learn the matching-to-sample rule. Then, several unfamiliar objects were added, and a generalization test was conducted to confirm that the subject could apply the matching-to-sample rule to the novel objects. After the test, several geometric patterns were added, and the subject was trained to apply the rule to those patterns that were visually similar to each other. Lastly, we changed the visual stimuli from real objects to photographs and then to a presentation on a monitor. Moreover, another generalization test was conducted to confirm that the subject could also apply the rule to the novel objects in the monitor presentation. Furthermore, we conducted a simple experiment to examine the visual abilities of killer whales using new geometric patterns in order to confirm the usefulness of this method.

## 2. Materials and Methods

### 2.1. Subject

One male killer whale (*Orcinus orca*), named “Earth” (14 years old) housed at the Port of Nagoya Public Aquarium in Aichi Prefecture, Japan, was used as the subject of this study. This subject undertook two-choice tasks in which visual stimuli were presented in monitors through an underwater window. The experimental procedure for the killer whales was approved by the Port of Nagoya Public Aquarium committee, as well as the Animal Welfare and Care Committee of Center for the Evolutionary Origins of Human Behavior, Kyoto University (No. 2022-076).

### 2.2. Apparatus

The training and experiment were conducted in a pool (elliptical shape, with dimensions of 34 × 11 m and 9 m in depth) with an underwater acrylic window (square, 2 × 2 m) located at a five meters depth. The experiment was conducted in a situation where the subject went back and forth between one experimenter (E1), who stood at the poolside directly above the underwater window, and another experimenter (E2), who stood in front of the window so that his head position was 60 cm away from the acrylic glass. All the visual stimuli were presented through this underwater window and, during the first half of the training, a workbench covered with black cloth was placed right in front of the acrylic glass so that the subject could not look behind it and the experimenter’s movements could not be used as a hint for the subject’s choice. In the latter half of the training, the workbench was replaced by a 43-inch monitor (94 cm width × 53 cm height, I-O DATA, EX-LD4K432DB). The monitor was placed 40 cm away from the acrylic glass, and was connected to a laptop behind it, and it was controlled by a custom program written in Microsoft Visual Studio 2019 (Microsoft Cooperation, Redmond, WA, USA).

### 2.3. Procedure

All of the experimental events were controlled by two experimenters. One experimenter (E1) at poolside gave a hand sign as the start signal to the subject, and he/she then directed it to the underwater window. At the same time, another experimenter (E2) started the presentation of a sample stimulus and waited for the subject to approach the window. After confirming a touch with the tip of its rostrum to the sample through the window, E2 removed the sample, and then presented two comparison stimuli. These comparison stimuli were presented at the same distance from the location where the sample stimulus was presented, and the distance between these two was 40 cm. From above the workbench or the monitor, E2 visually checked whether the whale’s rostrum touch position made to the comparison stimulus was to the right or left compared to where it was touching when the sample stimulus was being presented, which was defined as the subject’s choice. Since all stimulus presentations were made through an underwater window, the subject could not touch them directly, so the indirect touch made through the acrylic glass was treated as the touch to the stimuli. When the subject chose the correct answer, E2 blew a whistle and finished the presentation of stimuli. Then, E2 communicated, through a wireless device, the subject’s responses to E1, who was waiting at poolside. Then, E1 gave one or two pieces of fish to the subject as a reward so that the size of the reward was given equally per trial. When the subject chose the wrong answer, the presentation of stimuli was finished with neither a whistle nor reward. This series of the flow from the start signal to the feedback was regarded as one trial. The interval between the trials was defined as the time from when the subject returned to the poolside and rested in front of E1 until the next start signal was given, which was set to approximately five seconds. If the subject left the poolside for any reason, E1 waited for the subject to return and only then gave the start signal to initiate a trial. If the experimenter judged that the subject’s motivation was reduced due to consecutive incorrect answers, the subject underwent a training event, such as the “fin swing”, unrelated to the experiment. After being confirmed as having performed the task under calm states, it was rewarded. This method was applied to maintain the subject’s motivation. One session consisted of 12 trials, and the subject received 1 to 4 sessions in a day, with an inter-session interval of at least 5 min.

## 3. Training

### 3.1. The Teaching of the Matching-to-Sample Rule with Three Real Objects

This killer whale had originally learned one behavior as husbandry training: When it received a hand sign from the trainer at the poolside, it went to the front of the underwater window and touched a target of a white ball on the end of a black stick with its rostrum through the underwater window and returned to the trainer at the poolside after hearing a whistle sound. In this study, training was started by using this behavior and replacing the target with three real objects (boot, fin, and net; Figure 1) that the subject would see in its daily life at the aquarium. After it learned to touch those objects through the underwater window, it was trained to learn the rule of matching-to-sample. The training procedure using real objects is shown in Figure 2. At first, the experimenter at the poolside (E1) gave a start signal to the subject and instructed the animal to go to the underwater window. The experimenter standing in front of the underwater window (E2) held a sample stimulus with both hands and waited for the subject to approach and touch it with its rostrum. After confirming the touch, the sample stimulus was moved to behind the workbench, completely hidden from the subject’s view. Behind the workbench, E2 held the sample stimulus in one hand and another object as the incorrect stimulus in the other hand. Then, both hands were held up above the workbench and remained still for one second to show these two comparison stimuli to the subject. Lastly, both hands were stuck out in front of the subject simultaneously and if the subject chose the same object as the sample stimulus from the two comparisons, it was regarded as the correct action and E2 blew the whistle. The delay between the disappearance of the sample stimulus and the appearance of the two comparisons was one second. After hearing the whistle, the subject returned to the poolside and was rewarded with fish by E1. When the subject chose the wrong answer, the presentation of stimuli was finished with neither whistle nor reward. 

As a result of the training, i.e., 896 trials in 43 days, the subject maintained a steady 80% correct response rate, and the animal was considered to have learned the delayed identity matching-to-sample rule for these three real objects. This criterion was decided by the experimenters, and it was based on the fact that it is significant if the subject had 20 corrects out of the previous 24 trials (binomial test, *p*-value = 0.0015), and on the experimenter’s judgment that an 80% correct rate was enough high to maintain the subject’s motivation. As such, the subject was moved on to the next step of the experiment. 

### 3.2. The Addition of New Real Objects and a Generalization Test

As the next step, several objects that were unfamiliar to the subject were added, and the animal was then trained to perform a matching-to-sample task in the same way as the previous three objects. A light blue box (Figure 1) was added as the fourth object, and this was then paired with the previous three objects. The combinations containing this novel object were presented randomly in the trained trials in which familiar object combinations were presented. After adding the new objects, the subject tended to choose the familiar objects as the correct answer, but gradually it became able to apply the matching-to-sample rule to these new objects as well. After 213 trials in 9 days, the subject reached the criterion of 80% correct response rate and was considered to have learned the rule on the new objects. In the same way, a duckboard was added as the fifth object to the next 187 trials, which were conducted over 8 days. Then, a brown disk was added as the sixth object over the next 265 trials, which were conducted over 11 days. 

We conducted a generalization test using the seventh and eighth objects, i.e., a cushion and a chair (Figure 1), to confirm that the subject was able to generalize the matching-to-sample rule to unfamiliar real objects that they had never seen. The total number of trials for this generalization test was 76 trials, consisting of 16 probe trials and 60 baseline trials, and they were presented in a random order. In the probe trials, the two new pair of objects (the cushion and chair) were presented. In the baseline trials, two of the six trained objects were randomly selected and presented as pairs. The results of the test revealed that the subject correctly identified the objects 13 times out of the 16 probe trials, which represented a statistically significant result (binomial test, *p*-value = 0.021). Therefore, it was confirmed that the subject was able to apply the matching-to-sample rule to a pair of new real objects that they had never seen before.

### 3.3. The Addition of Geometric Patterns

The eight real objects used in the previous training steps were selected as objects judged to be easily distinguishable from each other in terms of color, size, and shape. This was performed because it was assumed that multiple objects with high similarity would cause confusion in the training of the matching-to-sample rule. In this step, we added real stimuli with geometric patterns that have been used in the previous visual cognition studies of bottlenose dolphins [36]. Four types of geometric patterns (circle, X-shaped, triangle, and H-shaped in Figure 3) were created by the same PVC tubing and were colored with yellow paint; as such, these objects were incredibly similar to each other in terms of color, size, and thickness. These geometric patterns were paired with each other and never paired with the eight previously trained objects. The combinations of geometric patterns were presented randomly in the combinations of trained real objects.

As a result of the training, i.e., 1256 trials over 46 days, the subject became able to maintain a stable 80% correct response rate for these geometric patterns. Compared to the result of the previous generalization test, this subject required a relatively long period of training for these patterns. This result suggested that this subject had difficulty in generalizing the matching-to-sample rule for these geometric patterns similar to each other. Then, we moved the training onto the next step.

### 3.4. Changing the Stimuli from Real Objects to Photographs

As the next step toward the monitor presentation, the stimuli were changed from real objects to photographs. Four objects (a boot, fin, duckboard, and cushion) were selected as “realistic patterns”, and they were attached onto blue plastic cardboards (Figure 4). Similarly, four “geometric patterns” were attached onto black plastic cardboards (Figure 4). These plastic cardboard “backgrounds” were devised to make the contours of the objects easier to recognize via emphasizing contrast. The subject first experienced that all stimuli were switched to the real objects with background boards. In addition, a monitor was placed 40 cm away from the acrylic glass instead of a workbench, and the experimenter (E2) adjusted his movements so that the location of the stimuli overlapped with the location of the monitor presentation. Then, the photographs of these visual stimuli were printed on papers in the same size and attached to the boards with double-sided tapes (Figure 5). The switch from the real objects to the photographs was performed simultaneously for the sample and comparison stimuli. Therefore, the subject did not have to learn the cross-dimensional matching between the real object and the photographs. During this process, the background color of the realistic patterns was changed from blue to gray. 

As a result of training for 652 trials over 27 days, the subject maintained a stable 80% correct response rate toward these photographs of real objects, and it was considered to have become ready to change to the monitor presentation; thus, it was moved onto the next step.

### 3.5. Changing the Stimuli from Photographs to Monitor Presentation

In this step, the subject experienced that all stimuli were switched from the presentation of photographs to the presentation in the monitor simultaneously. E2 stood behind the monitor and looked at the subject from above the monitor. And the laptop controlling the visual stimuli was also placed behind the monitor (Figure 6), so the subject could not see the laptop nor the hand movements of the experimenter manipulating it. After the start signal, E2 waited for the subject to approach with the sample stimulus presented in the center of the monitor. After confirming the subject touching the sample stimulus, E2 manipulated the laptop to erase the sample stimulus and present two comparisons on the both sides of monitor. If the subject chose the same match stimulus as the sample, the monitor was turned into a gray background at the same time as the whistle was blown. All stimuli displayed in the monitor had the same size as the photographs, and they were presented on a white background to make it easier for the subject to extract the target areas from the screen.

As a result of training for 554 trials over 19 days, the subject was able to maintain a stable 80% correct response rate for the monitor presentation. After this, the color of the background was changed to gray, and the presentation was changed to a state in which only the target objects were presented (Figure 7).

### 3.6. The Generalization Test in the Monitor Presentation Stage

In this step, a second generalization test was conducted to check whether the subject could generalize the matching-to-sample rule to objects that they had never seen before in the monitor presentation stage. The total number of trials in this generalization test was 144 trials, consisting of 24 probe trials and 120 baseline trials, which were presented in a random order. Four novel stimuli were each prepared for the realistic patterns; cup, chair_2, umbrella and leaf (Figure 1), and geometric patterns; rectangle, U-shape, D-shape and double-triangle (Figure 3). The results of the test revealed that the subject was correct 20 times out of the 24 probe trials, which represents a statistically significant (binomial test, *p*-value = 0.0015) result. Therefore, it was confirmed that the subject was able to apply the matching-to-sample rule to the pair of new real objects that it had never seen before in the monitor presentation. 

## 4. Experiment

### 4.1. Introduction

Through the training, it was confirmed that the killer whale became able to perform the delayed identity matching-to-sample task on the visual stimuli displayed on the monitor through an underwater window. This method made it possible to conduct the matching-to-sample tasks for the killer whale as if a touch panel were being used. Then, we conducted a simple experiment to confirm that this method was fully useful so as to examine the visual abilities of killer whales. In this experiment, four new geometric patterns were introduced, as shown in Figure 8. Each pattern, called a “Normal image”, was created by combining six white squares of a 6.6 cm × 6.6 cm dimension. At the same time, their symmetrical counterparts, called a “Mirror image”, were also created. The experiment consisted of 768 trials, one third of which (256 trials) were probe trials in which the pairs of these new stimuli were presented. The remaining 512 trials were baseline trials in which the pairs of the trained stimuli that were used in previous training stages were presented. Furthermore, the 256 probe trials were divided into two conditions. In 192 of the trials, the subject was required to discriminate between one Normal image and another Normal image (Normal vs. Normal condition). On the other hand, in the remaining 64 trials, the subject was required to discriminate between one Normal image and its Mirror image (Normal vs. Mirror condition). In general, the pairs of a Normal image and its Mirror image played an important role in the investigation of visual phenomena such as mental rotation. Furthermore, it is known that these pairs were more difficult to discriminate between than pairs of Normal images because they have similar characteristics. Therefore, the purpose of this study was to reconfirm that the matching-to-sample rule could be applied to novel images that have never been experienced before, as well as to investigate whether there is a difference in the discrimination between Normal vs. Normal pairs and Normal vs. Mirror pairs. Lastly, the study was also conducted to confirm that this method is useful for investigating the visual abilities of killer whales specifically.

### 4.2. Result

Figure 9 shows the results of this experiment. In Normal vs. Normal conditions, the subject was correct 168 times out of 192 trials (a correct rate of 88%), and this was confirmed as statistically significantly correct via a binomial test (*p*-value < 0.001). On the other hand, in Normal vs. Mirror conditions, the subject was correct 48 times out of 64 trials (a correct rate of 75%), and this was confirmed as statistically significantly correct via a binomial test (*p*-value < 0.001). The difference in the correct rates between these two conditions was confirmed as statistically significant via a chi-square test (X-squared = 4.7802, df = 1, and *p*-value = 0.029). The detail results of the correct rates for each combination of images are shown in Table 1. No statistical analysis was performed due to the small number of trials conducted in each combination.

### 4.3. Discussion

Despite the high visual similarity in these novel patterns, the correct response rate for the stimuli exceeded 80%, and thus it replicated the results in the second generalization test that were initially found in the training phase. The killer whale was able to perform a matching-to-sample task for visual stimuli that were presented on a monitor through an underwater window. Furthermore, it was also observed that the correct response rate was significantly lower for pairs of one Normal image and a Mirror image over pairs of two Normal images. This is consistent with the results obtained in previous studies on non-human animals, such as pigeons [56,57], rhesus monkeys [58], bushbabies [59], and chimpanzees [60]. Thus, it is suggested that this method works well in examining the visual abilities of killer whales, and that it is as useful as the use of touch panels.

## 5. General Discussion

In this study, one killer whale was trained to perform a delayed identity matching-to-sample task. As a result of the training, the subject was able to perform the task for visual stimuli presented on a monitor through an underwater window as if it was a touch panel. In addition, a simple experiment using novel geometric patterns was conducted, and it was indicated that this method is sufficiently useful for investigating the visual cognitive abilities of killer whales. The advantages of using this method can be described in four parts.

First, this method makes it possible to conduct experiments to examine the visual abilities of subjects by using only an underwater window, a monitor, and a computer. These devices do not require special knowledge or special skills unlike the specialized devices used in previous studies, such as underwater touch panels [43,44] or echolocation visualization systems [45]. Another advantage of this method is that it is less invasive as all the visual stimuli are presented through an underwater window, thus eliminating the need to submerge objects in the pool where the subject resides. This method is able to present visual stimuli in water, and it also makes it possible to examine the visual abilities of subjects that lives in underwater. This method can also resolve the questions regarding the validity of previous studies [35,36], in which visual stimuli were presented in the air, such as at a poolside, despite the fact that the experiment was designed to examine the visual abilities of subjects that live underwater.

The second advantage is that the presentation of the visual stimuli is performed by a monitor and a controlling computer. Although real objects are needed for the preparation in the early stages of the training, once the system can be switched to the monitor presentation stage, there is no need to prepare real objects as new stimuli as they can be easily generated as images on a computer. By presenting the visual stimuli on a monitor that emits its own light, it is possible to eliminate the differences between them in the way the stimuli are seen and illuminated when viewed from different view angles (which are such due to the use of real objects). Furthermore, the control of the experiment is simplified by the fact that the stimuli are manipulated by the pressing of buttons. In this experiment, all of the procedures can be controlled with only two experimenters, whereby the first experimenter delivers the start signal and rewards the subject, and the second experimenter operates the laptop and blows the whistle. As such, the possibility of experimenter effects is reduced by the presentation of the visual stimuli not being conducted by the experimenters themselves. In this study, the control of visual stimuli presented on the monitor was performed with a single button on the laptop placed behind the monitor, and this manipulation by the experimenter never provided any cues to the subject. Thus, cueing due to unconscious behavior of the experimenter (e.g., presenting the correct stimulus slightly earlier) and cueing based on sound produced by moving real objects, which can be an important hint for these animals with good hearing abilities, can be completely eliminated. In addition, although not performed in this study, the method is also expected to enable more accurate measurements of a subject’s reaction time by measuring the time taken to press buttons. Furthermore, the possibility of using moving images as visual stimuli is expected to increase the flexibility of future studies.

The third advantage is that this method makes it easier to increase the number of subjects and to conduct interspecies comparisons. One of the problems in cognitive studies on cetaceans is the small number of participating individuals. This method makes it possible to perform nearly the same task on the same species, but which are kept in different aquariums, which helps to secure the number of participating individuals in experiments. This is a point that would be incredibly difficult to achieve if special devices and real objects sets were used. It is also assumed that this method can be applied to other cetaceans that are kept in aquariums with similar underwater windows. Thus, it makes it possible to conduct interspecies comparisons and to solve one of the problems in the cognitive studies on cetaceans, i.e., where the target species has been limited to bottlenose dolphins. Similar to the present study, there have been several attempts to apply computerized tasks that have been used mainly with the terrestrial mammals to underwater mammals [40,41,42,43,44,45,61]. The widespread use of these methods in cognitive studies of underwater mammals will make interspecies comparisons across a wide range of species easier not only with other underwater animals but also with terrestrial animals. These comparative studies will greatly aid our understanding of the interactions of environmental and phylogenetic factors on the evolution of cognitive abilities.

The final advantage of this method is its contribution to animal enrichment and educational aspects in aquariums. Performing cognitive and behavioral tasks on captive primates in both laboratories and zoos has been shown to have a positive impact on their behavior [62,63,64,65]. Recently, touch panel systems have been shown to provide enrichment effects for primate and carnivore species [18,66]. A previous study has also shown that visual media through an underwater window could be an enrichment factor for killer whales [67]. Indeed, our experience in this study when using this method with a monitor showed that the subject was engaged in the sessions with over a 95% participation rate; in addition, the sessions were seldom canceled due to a lack of subject motivation. Furthermore, most zoos and aquariums have public education as one of their main missions and demonstrations aimed at educating visitors about animals have recently become increasingly popular. The impact of these demonstrations on visitors was such that they spent more time in front of an exhibition [68] and that they enjoyed it more [69]. In addition, they were more aware of the conservation efforts in zoos and aquariums [69], and they were more willing to donate to conservation projects [70]. The sight of a killer whale being assessed via an underwater window is surely of great interest to visitors, and it is expected to trigger their interest in the behavior and cognition of animals. As such, by presenting these benefits as compensation for the aquariums’ research collaboration, it is hoped that it will facilitate better understanding and cooperation with the cognitive studies in aquariums, thus leading to the establishment of a long-term cooperative relationship.

Although the advantages of using this method were described in the four parts, several problems can be raised regarding the fact that this method is not completely a touch panel task because it only uses an underwater window as if a touch panel were being used. First, this subject did not make the choice in direct touch to the visual stimulus, but rather the choice behavior was through the underwater window. In addition, another concern was that the final decision as to which stimuli the subject chose was made by the human experimenter. The response of the subject always included the time for the experimenter to determine which was the subject’s choice. Although there was the sufficient distance between the two comparison stimuli and whether the subject’s head moved to the right or left was enough clear behavior for the experimenter to judge the subject’s choice, we cannot completely rule out the possibility of an experimenter effect compared to the fully automated touch panel task. These concerns might be addressed by using a pressure-sensitive sensor that measures the impact of a subject’s touch of the window. In the future, there should be a constant need to improve this experimental method to solve these issues.

## 6. Conclusions

We succeeded in training one killer whale to perform a delayed identity matching-to-sample task for visual stimuli that were presented in a monitor through an underwater window as if a touch panel were being used. Through one simple experiment, this method was confirmed to be a useful tool for examining the visual abilities of killer whales. The use of this method solves some of the problems in previous visual cognitive studies on cetaceans and suggests the possibility of future comparative studies with other species. It is also expected to contribute to animal welfare in terms of cognitive enrichment, as well as for proposals for new exhibition methods in zoos and aquariums.

## Figures and Tables

**Figure 1 animals-14-00821-f001:**
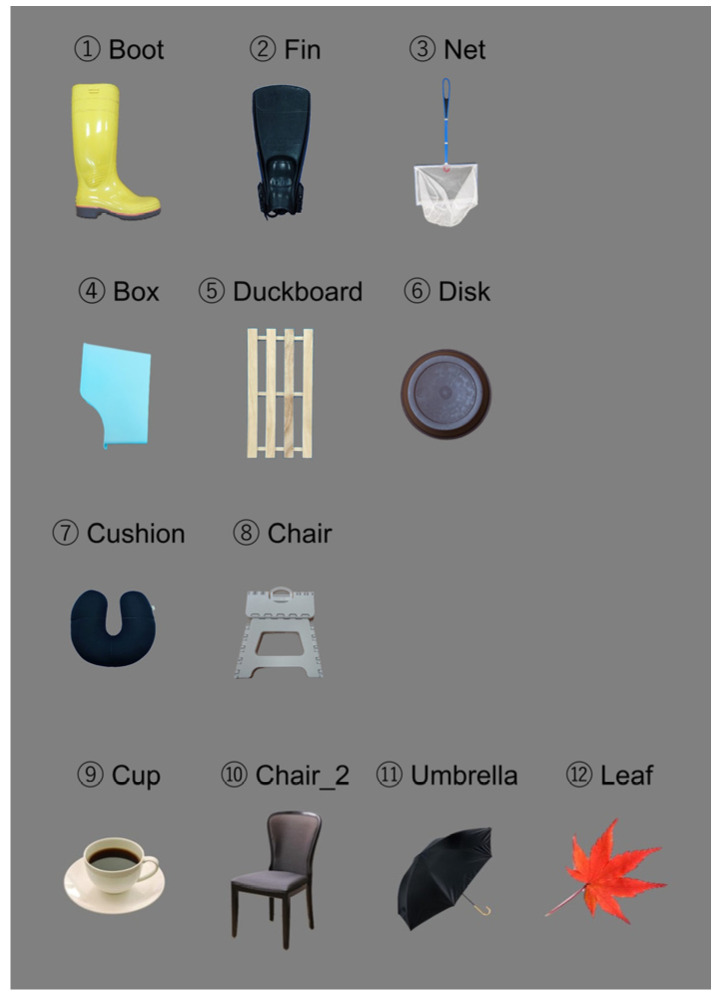
Objects used in the training.

**Figure 2 animals-14-00821-f002:**
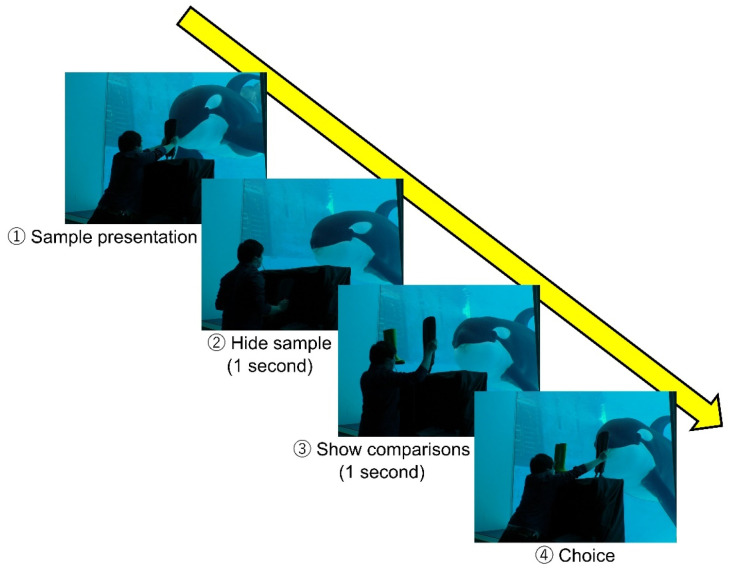
The procedure of one of the trials in the training using real objects.

**Figure 3 animals-14-00821-f003:**
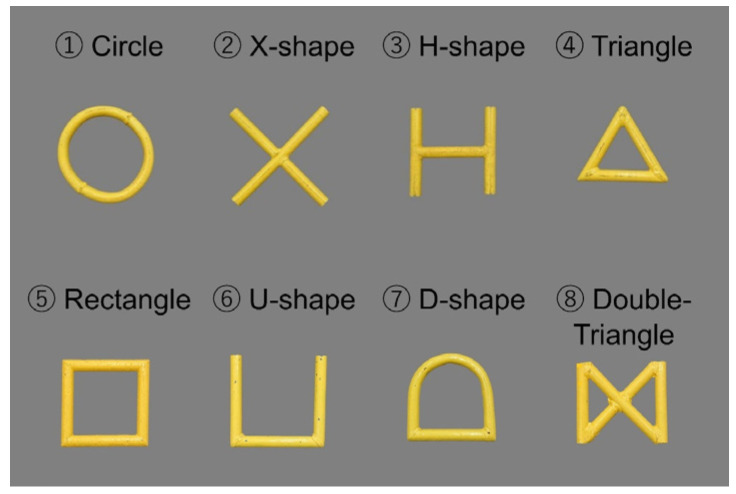
The geometric patterns used in the training. These patterns were previously used in the previous matching-to-sample task on bottlenose dolphins [36].

**Figure 4 animals-14-00821-f004:**
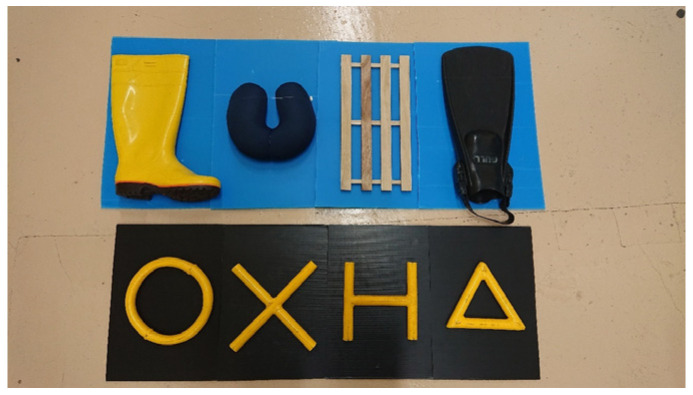
Four “realistic patterns” attached onto blue plastic cardboards and four “geometric patterns” attached onto black plastic cardboards.

**Figure 5 animals-14-00821-f005:**
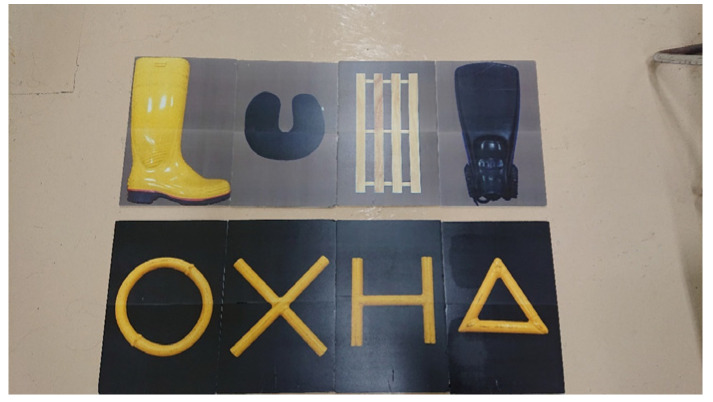
The photographs printed on paper and attached to the boards.

**Figure 6 animals-14-00821-f006:**
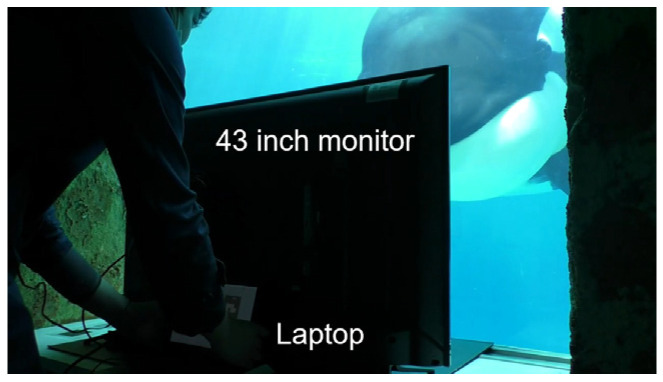
The experimental apparatus. A 43-inch monitor was placed in front of the underwater window, and it was controlled by a laptop that was placed behind.

**Figure 7 animals-14-00821-f007:**
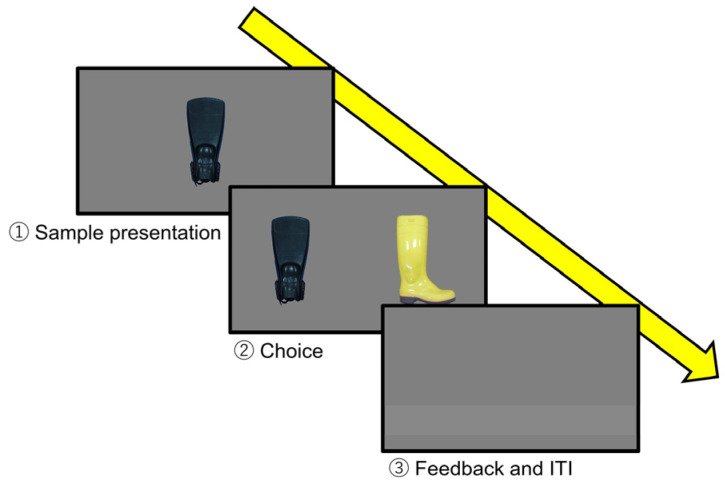
The procedure of one of the trials in the monitor presentation stage.

**Figure 8 animals-14-00821-f008:**
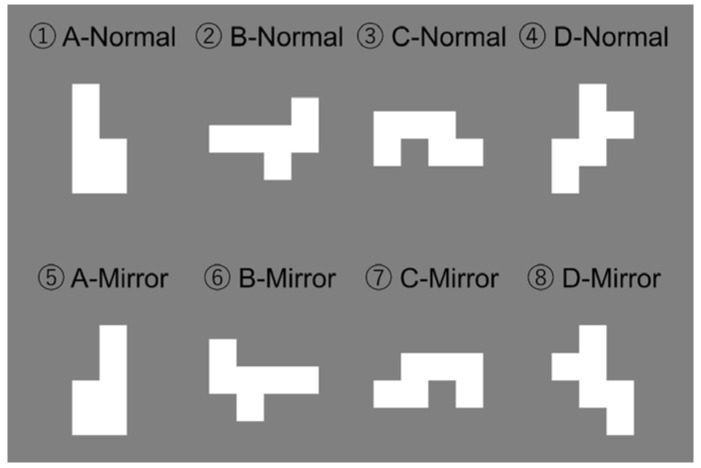
The geometric patterns used in the experiment. A total of four Normal images and four Mirror images were prepared.

**Figure 9 animals-14-00821-f009:**
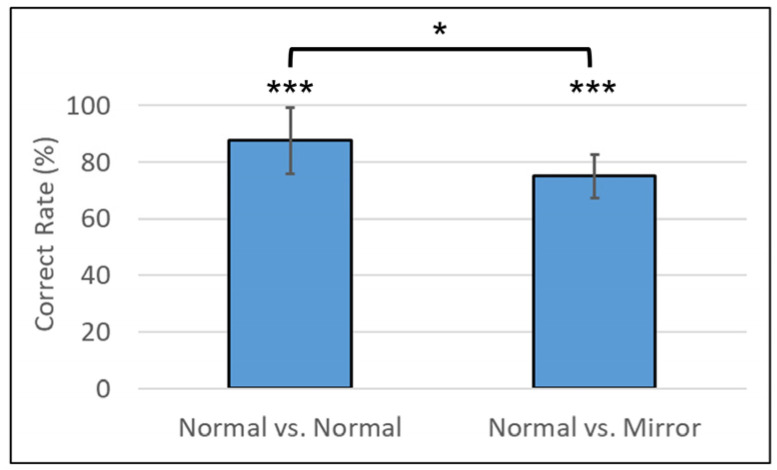
Mean percentages of correct rate (%) in experiment: Normal vs. Normal condition (Left) and Normal vs. Mirror condition (Right). The error bars show the standard error. Asterisks placed above each bar indicate the result of the binomial test, and an asterisk placed between two bars indicates the result of chi-square test (*** *p* < 0.001; * *p* < 0.05).

**Table 1 animals-14-00821-t001:** Detailed results of the correct rates (%) for each combination of images in experiment. In the result of Normal vs. Normal condition (Left), each slot indicates what percentage the subject selected the correct image in the combination of Normal and another Normal images. In the result of Normal vs. Mirror condition (Right), each slot indicates what percentage the subject selected the correct image in the combination of Normal and Mirror images.

**Normal vs. Normal**		Incorrect Image				**Total**	**Normal vs. Mirror**	
		A	B	C	D			
Correct Image	A		88	63	69	73	A	69
	B	100		88	94	94	B	69
	C	100	81		81	88	C	75
	D	100	94	94		96	D	88
Total		100	88	81	81	88	Total	75

## Data Availability

Data are contained within the article.

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
