# Peer review of "Matching-to-Sample Task Training of a Killer Whale (Orcinus orca)"

_animals, 2024, doi:10.3390/ani14060821_

Round 1

Reviewer 1 Report

Comments and Suggestions for Authors

Matching-to-sample task training of a killer whale (Orcinus orca) by Santa et al.

In this study, the authors sought to address several issues in the study the visuo-cognitive abilities of non-terrestrial animals, in particular cetaceans. One issue is lack of subject diversity (as most cetacean cognitive research focuses on dolphins). The other concerns the method of stimulus presentation to the animals, as stimuli are often presented in the air, even though these animals predominantly use their vision underwater.

Additionally, while touchscreens are commonly used in cognitive studies with terrestrial animals, it is difficult and expensive to develop for underwater use. In order to address these issues, the authors developed a touchscreen-like apparatus to assess the ability of an orca whale to learn matching-to-sample (MTS) rules and engage in mental rotation. They were able to train the orca to correctly complete matches with familiar real-life objects, generalize the MTS rule to unfamiliar objects, to complete matches with shapes, with physical photographs of the real-life objects and shapes, with photographs of both stimulus types presented on a monitor, and then to complete matches with geometric shapes and their mirrors. They found that the orca was more accurate on normal matches than on mirror matches, a pattern that has been seen in other animals. Altogether, this study has a large number of implications for the methodology used to study aquatic animals, the breadth of species studied, public education and engagement with cognitive science conducted in zoos and aquariums, and the welfare of animals used in cognitive testing.

Major concerns:

There is a lot of important information missing from the methodology section in particular, as detailed below (and also in terms of some of the minor issues noted).

For instance, what experimental procedures were used to prevent experimenter cueing, particularly by E2 in the real-life object and photograph training phases? If we are to assume these animals could see the stimuli through the glass, couldn’t they also potentially pick up on body language cues from the experimenters holding the stimuli and providing feedback? You made a good point in the Introduction that computers remove the human, but this was not true here, so did you try to control cuing or just not worry about that given the ultimate aim to move to the computer? The General Discussion also highlights this limitation.  This is a good thing to include, as well as the future additions to the system such as touch-pressure monitoring that could make this a more objective data collection technique.

More detail is needed in the Apparatus section about the location of the stimuli/screen. How far was it from the acrylic window? How close was the second experimenter usually to the window? As you mentioned in the discussion, the fact that the subject was discriminating objects through not only water but also the plexiglass and potentially air between the stimuli and the window, it may be important to know the physical location of stimuli.

Was the subject previously trained to touch targets at the window, or did this behavior need to be shaped?

What was the reasoning for the 80% performance criterion signifying that the animal had learned the MTS rule? Is it based on prior research, or was this determined by the researchers? Additionally, what was used to calculate this performance (e.g., percent correct in the previous # trials)?

In section 3.4 Changing the stimuli from real objects to photographs, it’s a bit unclear how the stimuli were presented. Specifically, was the sample stimulus in these trials the real-life object and the matches were photographs, or vice versa, or were all stimuli used in this phase photographs? It would be important to know whether the orca was learning to match across (e.g., 3D > 2D) or within (2D > 2D) dimensions.

When the screen was introduced, the human experimenter still determined what was “touched” since there was no actual contact to the screen, correct?  How does this prevent the cuing concerns that were raised earlier in the paper? 

Without being fully automated, it is not clear what presenting stimuli on a monitor screen versus a real screen or board adds to our ability to assess behavior and cognition in animals that are underwater?  Is it mainly the complexity of stimuli that can be used, and the ease of being able to more quickly prepare trials? 

Minor concerns:

Line 28: change “performs” to “performed”

Lines 84-86: Is there any literature on this? It’s a good question.

Line 105: change “sizes” to “size”

Lines 119-120: this is confusing.  Was the sample present at the same time as the match choices?  If so, this is called simultaneous matching to sample.  If it was removed before the match images appeared, this is called delayed matching to sample.  Please clarify.

Line 120: change to “The participant received positive feedback if it chose one of the stimuli that was the same as the sample.”

Line 121: Change “can receive” to “received”.

Lines 125-127: The phrasing of this is a bit confusing. I think I know what is meant, but you might want to rephrase.

Lines 130-133 are redundant with earlier information.  Those should be removed.

Line 138: How old was the subject? Matching-to-sample learning and performance has been shown to be influenced by age in humans and macaques (Paule et al., 1998), and rats (Dunnett et al., 1988), so it may be important to note the age of this subject, especially if he was very young or very old.

Line 138: Change “and who is housed” to “housed”

Line 160: What was the start signal?

Line 160: change “it” to “the whale” or something similar

Line 163: change to “after confirming a touch with the tip…”

Lines 162-164: how exactly was this done?  It is not clear how the trial progressed, and more detail is needed.

Line 167: When was the subject awarded one vs. two pieces of fish? Was the goal to make sure all rewards were the same size (e.g., two small pieces were equal in size/weight to one big piece)?

How did the participant choose a match response?  This was not clear?  How did the experimenter decide when a choice was made? 

Did the whale swim back to E1 after making a response?  Was this trained?

Line 179: call this an “inter-session interval”

When you say “touch it with its rostrum” you do not mean physically touch it, correct, since it was on the other side of a window?  You mean attempt to touch it through the window, right?

Line 198: “and was rewarded”

Lines 214-217: I might remove “was puzzled by them and”.

Lines 216-221: it would have been good to have a criterion set to see whether this was faster and faster acquisition of the criterion as the set got larger.  Is there any way you can assess that, as it would be stronger evidence that the matching rule was being established independent of specific stimulus features.

Line 225: You might want to change the terminology from “trained trials” and “test trials”, as it’s a bit unclear what is meant by this. Maybe something like “probe trials” for the new items?

Line 307: “Experiment” instead of “The experiment”.

Table 1: This table is a bit difficult to interpret, and it may be clearer if you describe results in a graph. Assuming no statistical difference between the image pairs in the Normal vs. Normal condition (which you should report in the results section regardless), you could have a bar for performance in the two conditions, or a bar for each image (A, B, C, D) within each condition. Something to make this data more visual could be more accessible to readers.

Line 351: This relationship should be reversed – the results here (in the Experimental phase) replicated the results in the second generalization phase.

Comments on the Quality of English Language

No concerns.

Author Response

Thank you very much for taking the time to review this manuscript. 

Reviewer 2 Report

Comments and Suggestions for Authors

Please review the comments on the manuscript.

A few of the concerns regarding the framework used to establish the background for the study, needed clarification on test trials in context of training, and general editing of statements that do not accurately represent the sensory abilities and our knowledge. Would recommend emphasizing visual abilities and cognitive studies that have been conducted with killer whales - imitation, viewing tv monitors for self-awareness and enrichment, and other studies rather than comparing and contrasting the acoustic/echolocation abilities of the killer whales. The study does not use this modality and should not be half the introduction.

Comments on the Quality of English Language

Quality of English was generally fine. Identified a few areas that could be clarified or different word choices.

Author Response

(The authors gave the same response as above.)

Round 2

Reviewer 1 Report

Comments and Suggestions for Authors

The authors have improved the paper substantially. They have tried to address all of the concerns raised on the first review.  The paper is closer now to being potentially published, but there are a few things that would still need to be added/revised before formal acceptance is recommended.  Most of these are wording changes to clarify meaning, but some of these comments also need more substantive responses by the authors.

Lines 119-120 – this needs to be rephrased.  Perhaps you mean “the small amount of research to examine the visual cognitive…”

Line 157 – “experimenter’s movements could not be used…”

Line 167 – “Then, after confirming a touch with the tip…”  delete “the”

Line 168 – here, to confirm, this is delayed matching, right?  If so, maybe say “the sample was removed, and two comparison stimuli then were presented.”  This would clarify this is a DMTS task.

Line 171 – delete first word “of”

Line 171 – “whether the whale’s rostrum…”

Line 195 – delete “one of”

Line 221 – “subject” rather than subjects”

Line 269 – “had difficulty”

Line 309-310 “on both sides of the monitor” – delete “the”

Line 310 – “If the subject chose the same match stimulus as the sample,…”

Figure 9 – with only one subject, what means are being shown here?  You should present the overall percent correct, and statistically analyze with a binomial test, as you have been doing in other places.

Table 1 should start – “Detailed results…”

Lines 389-392 – this is a long and difficult sentence.  Recommend breaking this up.  What about “Despite the high visual similarity in these novel patterns, the correct response rate for the stimuli exceeded 80%, and thus it replicated the results in the second generalization test in the training phase. The killer whale was able to perform a matching-to-sample task for visual stimuli that were presented on a monitor through an underwater window.”

Comments on the Quality of English Language

Minor edits needed, as I noted in my specific comments.

Author Response

Thank you very much for the time to review this manuscript.

Please find the detailed responses in the attached file and the resubmitted manuscript.

Reviewer 2 Report

Comments and Suggestions for Authors

The author did a nice job responding to the comments and suggestions. Aside from some clean up of grammar and phrasing due to English being a second language, the paper is ready to go.

Comments on the Quality of English Language

A few minor areas for clarifying English phrasing/grammar. Expand contractions.

Author Response

Thank you very much for taking the time to review this manuscript.

We would like to thank you for your careful pointing and helpful suggestions, which helped us to improve our manuscript significantly.